# Spearmint (*Mentha spicata* L.) Phytochemical Profile: Impact of Pre/Post-Harvest Processing and Extractive Recovery

**DOI:** 10.3390/molecules27072243

**Published:** 2022-03-30

**Authors:** Karina Sierra, Laura Naranjo, Luis Carrillo-Hormaza, German Franco, Edison Osorio

**Affiliations:** 1Grupo de Investigación en Sustancias Bioactivas GISB, Facultad de Ciencias Farmacéuticas y Alimentarias, Universidad de Antioquia, Calle 70 No. 52-21, Medellín 0500100, Colombia; karina.sierra@udea.edu.co (K.S.); meliza.naranjo@udea.edu.co (L.N.); lcarrillo@bioingred.co (L.C.-H.); 2Bioingred Tech S.A.S., Tech Innovation Group, Calle 46 No. 41-69, Itagüí 055412, Colombia; 3Corporación Colombiana de Investigación Agropecuaria Agrosavia, Centro de Investigación La Selva, Rionegro, Llanogrande 054048, Colombia; gfranco@agrosavia.co

**Keywords:** *Mentha spicata*, phenolic compounds, carvone, essential oil, rosmarinic acid

## Abstract

The purpose of this study was to chemically compare samples of *Mentha spicata* (marketing byproducts, production byproducts, and export material), cultivated in the open field and under greenhouse, using an integrated approach by HPLC/DAD and GC/MS analysis. The presence of phenolic compounds was higher in the marketing byproducts cultivated in the open field. Marketing byproducts also had the highest amount of carvone. For this reason, this byproduct was selected as a candidate for the development of natural ingredients. With the best selected material, the optimization of simultaneous high-intensity ultrasound-assisted extraction processes was proposed for the recovery of the compounds of interest. This extraction was defined by Peleg’s equation and polynomial regression analysis. Modeling showed that the factors amplitude, time, and solvent were found to be significant in the recovery process (*p* < 0.005). The maximum amount of compounds was obtained using 90% amplitude for 5 min and ethanol/water mixture (80:20) for extraction to simultaneously obtain phenolic and terpenoid compounds. This system obtained the highest amount of monoterpenoid and sesquiterpenoid compounds from the essential oil of *M. spicata* (64.93% vs. 84.55%). Thus, with an efficient and eco-friendly method, it was possible to optimize the extraction of compounds in *M. spicata* as a starting point for the use of its byproducts.

## 1. Introduction

The demand for products made from medicinal plants has increased considerably in terms of market and exports. However, there is also a negative environmental impact due to the generation of byproducts, mainly solid wastes, during the production processes exploiting roots, stems, seeds, flowers, and fruits, with few alternatives for the use of these materials. Thus, many of the discards become sources of contamination [1]. This situation has motivated interest in the use of byproducts in different areas, mainly the search for new technologies that use byproducts in the production of raw materials with added value, contributing to the reduction in costs and the reduction in the environmental impact [2]. In the specific case of *Mentha spicata* L. (spearmint), the current market and its imminent growth represent problems of byproduct generation throughout the production chain, which means that not all the material collected is marketed in its entirety, generating wastes of approximately 40% [3]. Byproducts can occur at all stages of production: harvest, postharvest, processing, and distribution stages, either for environmental or animal problems, but mainly with the selection of material that does not meet quality parameters required in the export processes (marketing byproducts) [3,4].

Spearmint is a species belonging to the Lamiaceae family, which has approximately 7285 species in 250 genres distributed throughout the world, mainly in the Mediterranean [5]. Specifically, *Mentha* has eighteen species, thirty-one subspecies or varieties, of which approximately twelve are natural hybrids [6]. *M. spicata* is widely used for its essential oil in many industries. The main component of essential oil is carvone (CA), a monoterpenoid compound [7]. Moreover, the main phenolic compound corresponds to rosmarinic acid (RA), which is a powerful antioxidant compound [8]. In an effort to take advantage of the compounds of interest present in spearmint, alternatives to traditional extraction methods are being sought, as the traditional extraction methods are time-consuming, inefficient, nonspecific, and require a high consumption of organic solvents [9]. For this reason, new trends in extraction processes are expected to be more related to green chemistry, with high extraction yields, allowing minimization of costs and processing times, with lower energy consumption and vapor emissions [10]. These extractions are known as green extractions or modern extraction techniques, which present cost–benefit modifications that reduce the problems [9]. The trends focus on technologies such as supercritical fluid extraction (SFE), microwave-assisted extraction (MAE), and high-intensity ultrasound (HIU) extraction [11]. The main interest in the latter is due to its high impact for contaminant-free extractions, high yields, specificity, and/or simultaneity toward compounds of interest and scalability [12,13].

The effects caused by HIU are related to the phenomenon known as cavitation, which produces a breaking of the cell walls due to their compression and decompression [13,14]. In this way, the penetration of the solvent and the extraction yields is favored [15]. The boom of HIU has increased worldwide due to its application in the industry of phytotherapeutic products [14]. This technique is seen as an interesting tool for recovering compounds from plant sources, as it is a clean, safe, and energy efficient technique [9]. HIU allows the reduction in production costs and facilitates the scaling of the process, which is more difficult with other extraction methods [16]. The present work was conducted to compare the content of phenolic and terpenoid compounds between samples of *M. spicata* from different byproducts obtained from crops cultivated in the open field or under greenhouse conditions using an integrated approach by high performance liquid chromatography/diode-array detection (HPLC/DAD) and gas chromatography/mass spectrometry (GC/MS) analysis. The best plant material was selected, and an optimization of high intensity, ultrasound-assisted extraction process was proposed for the recovery of compounds of interest. In this way, it is intended to contribute to the development of ingredients for different industrial specialties at the pharmaceutical, chemical, and food levels.

## 2. Results

### 2.1. Selection of Extraction Solvent

The extraction conditions for the different analyses were previously chosen in the laboratory, where plant material, extraction time, and solid/solvent percentage were determined as follows: leaves, 30 min of extraction and 16% plant material, respectively (Appendix A). Then, the impact of the extraction solvent was determined. For this purpose, 6 mixtures were established with different proportions of ethanol in hydroalcoholic solutions from 50% to 100% (%*v/v*). The analysis was carried out with the marketing byproducts cultivated in the open field, because it was the sample with the largest amount of plant material available. Six extracts obtained were analyzed for their TPC and RA determined by HPLC/DAD. The 80% ethanol solution presented the best extraction result in terms of the compounds of interest (Figure 1A,B). This behavior was observed for both polyphenols and RA. In general, an increase in extraction performance was evidenced when the percentage of ethanol increased to 80%; thereafter, the extraction values started to decrease. However, no significant differences were obtained among the 60%, 70%, and 90% ethanol mixtures for the content of RA. While there were no differences in TPC values of 60%, 70%, and 80%, the last mixture was selected as the extractant solution to perform the tests in this work. In addition to the evaluation of phenolic compounds, terpenoids commonly present in the essential oil of *M. spicata* were also extracted with the chosen solvent, as the objective was related to the possibility of taking advantage of this type of compound obtained from the residues of spearmint. However, it is known that not all constituents of essential oils are polar, and ethanol could partially extract some compounds. For this reason, the chemical composition of the essential oil was analyzed, and the degree of extraction of volatile compounds that can be achieved using solvents with different polarities, thanks to its ease of dissolving nonwater-soluble substances, was determined [17]. The solvents evaluated were hexane, dichloromethane, and ethyl acetate; the proportions were hexane:ethyl acetate 50:50 and 75:25; and the solvent chosen in the previous section was 80% ethanol. The results obtained show that the highest amount of terpenic compounds (monoterpenoids and sesquiterpenoids) were obtained with 80% ethanol (Table 1). Therefore, ethanol allows the extraction of both nonpolar and polar compounds from spearmint. Once the analysis conditions were determined, the export and byproduct samples of spearmint were compared in terms of TPC, RA, and carvone.

### 2.2. Determination of Total Phenolic Content on Byproducts and Export Material

The TPC value was determined in the analyzed spearmint samples. The total amount of polyphenols in the samples ranged from 6.40 to 9.26 mg GAE/g extract (Figure 2A), where marketing byproducts had a significant content of phenolic compounds in relation to other samples. The comparative analysis between the different growth conditions indicates that the samples that presented a higher TPC value correspond to those cultivated in the open field, presenting significant differences (*p* value < 0.05) with the samples cultivated under the greenhouse conditions (Figure 2B).

### 2.3. Phenolic Compound Identification by HPLC-DAD on Byproducts and Export Material

The presence of RA was determined in the extracts using HPLC-DAD at 390 nm. In the comparison between byproducts and export material (Figure 2A), the latter was observed to have the highest amount of RA. In addition, the samples with the greatest number of phenolic compounds, as well as RA, correspond to the plants cultivated in the open field with respect to those cultivated under the greenhouse conditions (Figure 2C). Other compounds were identified by comparison of ultraviolet (UV) spectral and retention times with reference compounds. These compounds were chlorogenic acid, caffeic acid, kaempferol, and luteolin-7-*O*-glucoside, but it was not possible to quantify their lower content in the sample.

### 2.4. HPLC-DAD-ESI/MS^n^ Analysis on Marketing Byproducts

To obtain more information on the phenolic compounds present in marketing byproducts, the samples with higher content of phenolic compounds, was performed an identification analysis using HPLC-DAD-ESI/MSn. Twenty-seven compounds were tentatively identified in spearmint extract based on retention time, mass spectra and fragmentation patterns found in databases created in the laboratory for phenolic compounds (Table 2). In addition, these flavonoids and phenolic acids were identified based on the analysis of their characteristic mass spectra (MS2 and MS3) and their comparison with the information reported in the literature. Furthermore, RA (m/z 359, MSn 359, 223, 197, 161) is the major phenolic compound, and the detection of different derivatives of RA was also possible.

### 2.5. Gas Chromatography/Mass Spectrometry (GC/MS) Analysis on Byproducts and Export Material

Once the extraction solvent ethanol:water 80:20 was defined, the comparison of export material and byproducts in relation to terpenoids commonly present in the essential oil of spearmint was also performed (Table 3). The results show that carvone, trans-carveol, β-caryophyllene, and germanecrene D correspond to the majority of the extracted compounds. Extraction with solvents was found to allow up to 30% of the terpenoids present in the essential oil to be obtained. Subsequently, the majority of the compounds were quantified using a calibration curve. Carvone content was present more often in marketing byproducts than in export material or production byproducts (Figure 2D and Table 3). However, the total relative percentage of terpenoids was similar between export material and marketing byproducts (Table 3).

### 2.6. Kinetics of Solid–Liquid Extraction by High-Intensity Ultrasound on Marketing Byproducts

The extraction was carried out with 7% of plant material and 80% ethanol as extraction solvents, and both conditions were optimal in previous tests (Appendix A). Therefore, the influence of time and ultrasonic amplitude on ultrasound-assisted extraction was evaluated. In the case of the kinetics of RA, an adequate fit of the experimental data was obtained with the model applied with an R^2^ of 0.99 (Figure 3A). According to the extraction kinetics, differences were found in the amplitudes evaluated, although the extraction curves obtained showed a similar behavior for TPC with an R^2^ of 0.99 and carvone with an R^2^ of 0.96 (Figure 3C,D). The concentration of RA increased substantially in the first 5 min with a growth ramp up to the 10th minute; after this time, the increase in concentration was reduced until reaching a plateau (Figure 3A). These values were directly proportional to the amplitude used, which increases the temperature in the system (Figure 3B). An increase in temperature is related to the increase in solubility by solvent saturation [18]. With the results obtained, an amplitude of 90% and a time of 5 min were chosen. Carvone (Figure 3C) and TPC (Figure 3D) results have a behavior similar to the behavior observed with RA, where the influence of amplitude favors their extraction.

A polynomial regression model was analyzed using the same information of Peleg’s kinetics (time and amplitude) using response surface methodology (RSM). The results are shown in Figure 4. According to the results, the effects of the variable extraction time (X_1_), amplitude (X_2_), interaction X_1_X_2_, and quadratic interaction X_1_^2^ and X_2_^2^ were statistically significant (*p* value < 0.05) for all analyses (Appendix A). The R^2^ value was 0.92 for the TPC analysis, and similar values were obtained for the RA and carvone analyses, where the R^2^ values were 0.91 and 0.99, respectively. These results indicate the correlation of predicted values. In general, Peleg’s equation explained the extraction in the compounds of interest in a more adequate way, because it described the equilibrium concentration. Instead, the polynomial regression provides a quadratic curve that was not observed in the experimental data. For this reason, the values chosen in the extraction of the metabolites of interest were an amplitude of 90% and a time of 5 min.

## 3. Discussion

The chemical characterization of *M. spicata* samples (marketing/productions byproducts and export material) allows the determination of the potential of the species for the possible development of a natural ingredient of interest in the pharmaceutical or food industry. With respect to the selection of solvents for the development of extracts, an extraction mixture consisting of 80% ethanol and 20% water was found to favor the joint extraction of phenolic and terpenoid compounds. In this way, the functional properties and organoleptic characteristics of spearmint can be jointly exploited. This information is of great relevance as, with the same extraction model, phenolic and terpenoid compounds were found to be obtained [19], resulting in a sustainable process that requires less time and uses ecological solvents that combine a high yield in obtaining compounds of interest. Ethanol was the only extraction solvent used because ethanol is an approved solvent in the food industry, and it also presents higher extraction efficiencies [20,21].

Many phenolic compounds have been demonstrated to have various properties, including antioxidant activity [22]. Therefore, it is not surprising that extracts of *M. spicata* are frequently used as preservative agents that decrease oxidative peroxidation during food storage [23]. The samples that presented a higher TPC value correspond to those cultivated in the open field. In general, some studies report that the level of phenolics is higher in plants exposed to light than in samples from plants covered with nets and is highly influenced by the environmental conditions [24,25]. The accumulation of phenolic compounds in plant tissues is a distinctive feature of environmental stress, and an increase in the biosynthesis of polyphenolic compounds helps plants cope with multiple biotic and abiotic stresses, such as drought, heavy metals, salinity, temperature, ultraviolet light, etc. [26,27], which could explain part of the results obtained.

Chemical analysis was used to determine the phenolic acid, flavonoid and terpenoid compounds in samples of *M. spicata*. The chemical study by HPLC/DAD showed that rosmarinic acid is the major phenolic compound of *M. spicata*, which was verified by HPLC/MS [28]. RA has several food applications and health-beneficial properties [29]. Rosmarinic acid is distributed in the family Lamiaceae and specifically in the species of interest, presumably accumulating as a defensive compound [30]. However, RA content rarely exceeds one percent per dry weight of the material [31] and depends on different factors related to the growth, the environment, and the crop conditions [29]. In addition, flavonoids and phenolic acids were also identified. Some flavones (derivates of luteolin and apigenin), flavanols (derivates of kaempferol) and flavanones (narinrutin) were identified in the spearmint extract [32], and the detection of different derivatives of RA was also possible. All compounds present in the sample (Table 2) were previously reported for *M. spicata* [22,33,34,35]. On the other hand, carvone was determined to be the major terpene constituent analyzed by GC/MS. This compound was present more often in marketing byproducts than in export material or production byproducts (Figure 2D and Table 3). With this information, the marketing byproducts are presented as candidates for the development of natural ingredients.

Subsequently, a method of simultaneous extraction assisted by high intensity ultrasound and evaluation by HPLC/DAD and GC/MS was developed for the study of bioactive compounds. HIU can be performed with an ultrasonic generator probe or with a sonicator bath. The ultrasonic probe is more powerful due to the surface distribution of the ultrasonic intensity [10]. Therefore, this system was selected for the extraction process. In the present study, the collected data were processed through mathematical model that facilitates optimization, simulation, design, and control in the process that improves the use of energy, time, and solvent [36]. The Peleg equation is an empirical and classical hyperbolic model for describing moisture absorption curves. Due to the resemblance between sorption kinetics and extraction, the Peleg model has been adapted to describe the extraction kinetics of different plant compounds [37,38,39]. The physical parameters used in the HIU technique correspond to amplitude, frequency, power, and wavelength. Some of them are important to establish a scalable process, such as the amplitude [37]. In addition, some parameters are related to the apparent viscosity of the system, such as time extraction and the percentage of plant material. Therefore, the influence of time and ultrasonic amplitude on ultrasound-assisted extraction was evaluated. Through the design of experiments, it was able to standardize an extraction process using high-intensity ultrasound equipment to maximize the extraction of the compounds of interest present in *M. spicata*, minimizing the use of conventional technologies that generate high-energy costs and are highly polluting. Thus, the extraction conditions of ultrasonic amplitude 90%, time of 5 min and a load of 7% of plant material for the extraction of the compounds of interest were established. In addition, a starting point for the development of valuable products is provided in view of the chemical and functional potentialities previously reported in spearmint.

## 4. Materials and Methods

### 4.1. Chemicals

All solvents were analytical and/or HPLC grade. The standard carvone was acquired from Sigma Aldrich (Buchs, Switzerland), and luteolin 7-*O*-glycoside was acquired from Sigma Aldrich (Saint-Quentin-Fallavier, France). Chlorogenic acid, caffeic acid, and luteolin were purchased from Sigma Aldrich (St. Louis, MO, USA). Rosmarinic acid was purchased from the European Pharmacopoeia reference standard (Europe). P-Coumaric acid and apigenin were purchased from Extrasynthase (Genay, France). Ferulic acid and hesperidin were purchased from Fluka (Buchs, Switzerland). Rutin and naringin were purchased from LKT Laboratories (St. Paul, MN, USA). All standards were prepared as stock solutions in ethanol (1 mg/mL).

### 4.2. Plant Extracts

*M. spicata* was obtained from three producing and exporting farms located in eastern Antioquia, Colombia, during the months of July and October 2020. In addition, plant material was also collected at Agrosavia’s La Selva Research Center. In all cases, the samples were grown in open fields and under greenhouse conditions. Once the harvested plant material is ready to be collected, *M. spicata* is cut and destined for export (export material). The material left in the field after cutting and material that has suffered physical or pest damage correspond to production byproducts. A last sample was taken from the residues produced at the commercialization companies in charge of export material (marketing byproducts). Therefore, 3 types of samples were analyzed: export material, production byproducts, and marketing byproducts; each obtained by 2 different growing conditions: open field and greenhouse; for a total of 6 experimental treatments (Appendix A). The leaves of *M. spicata* were separated, washed, disinfected, dried (45 °C for 72 h), and ground in an electric blender, and the powder was stored at room temperature (24 °C, 60% RH).

### 4.3. Preparation of Extractions

For the determination of the extraction solution, the total polyphenol content (TPC) assay and the quantification of rosmarinic acid by HPLC/DAD were performed. Previous conditions of time extraction and solid/solvent percentage were evaluated for the extracts. To obtain phenolic compounds, 5 extraction systems were selected with different ethanol/water ratios from 50:50 to 100:0 (*v/v*). Approximately 20 g of sample was mixed with 100 mL of the extraction solution, and the extraction process was performed with an ultrasonic bath (Elma P60H, Singer, La Vergne, TN, USA) at 37 Hz and 30 ± 5 °C for 30 min. The supernatants were filtered with filter paper, combined, and centrifuged at 13,000 rpm for 20 min (Sorvall, Thermo Scientific, Waltham, MA, USA) and stored at 4–7 °C until subsequent analysis. For GC/MS analysis, the essential oil was obtained by hydrodistillation for 5 h at 80 °C using a Clevenger-type apparatus, dried with anhydrous sodium sulfate, and stored at room temperature until subsequent analysis.

### 4.4. Determination of Total Phenolic Content (TPC)

The Folin–Ciocalteu method was carried out to determine the TPC content of each extract [40]. Gallic acid in a dynamic range of 10 to 100 μg/mL was used as a reference standard. Then, 25 μL of the extract was mixed with 125 μL of Folin–Ciocalteu’s reagent (1:10), both diluted in distilled water. The mixture was shaken and incubated in darkness for 5 min at room temperature, followed by the addition of 100 μL of Na_2_CO_3_ (7.5% *w/v*). After 60 min of incubation at room temperature in the dark, absorbance readings were performed at 765 nm using a Synergy HT multimodal microplate reader (Biotek Instruments, Inc.; Wonooski, VT, USA). The total polyphenol content was calculated using a calibration curve with gallic acid. The results are expressed as milligrams of gallic acid equivalents (GAE) per gram of dry sample or extract, as appropriate (mg GAE/g dry sample).

### 4.5. Phenolic Compound Identification by HPLC-DAD

HPLC analysis was carried out on an Agilent 1200 series instrument (Agilent Technologies, Palo Alto, CA, USA) equipped with a vacuum degasser, an automatic autosampler, a quaternary pump and a DAD. Compound separation was performed using a Zorbax SB RRTH^®^ (fast resolution and high throughput) C18 column (50 mm × 4.6 mm with 1.8 μm particle size) at 30 °C, with a flow rate of 1.0 mL/min. Once the separation methodology was optimized, the mobile phase consisted of 0.5% formic acid in water (A) and acetonitrile (B), and the linear gradient used was as follows: 0 min, 16% B; 4 min, 16% B; 8 min, 20% B; 11 min, 40% B; 12 min, 45% B; 13 min, 50% B; 14 min 60% B; and 15 min, 16% B. The injection volume was 5 μL. The analyzed compounds were monitored in the DAD at 250 and 329 nm. For the quantification of RA in the samples, the external standard calibration method was used.

### 4.6. HPLC-DAD-ESI/MSn Analysis

HPLC-DAD-electrospray ionization (ESI)/MS was used to analyze the ethanolic extracts of spearmint using an Agilent HPLC 1200 equipped with a photodiode array detector and a mass detector in series (Agilent Technologies, Waldbronn, Germany). An ion trap spectrometer (Model G2445A), which included an electrospray ionization interface and was controlled by LCMS software (Agilent, version 4.1), was used as a mass detector. The column was a Phenomenex Luna C18 250 × 4.6 mm with a 5 µm particle size. The mobile phase consisted of 0.1% formic acid in water (A) and acetonitrile (B), and the linear gradient used was as follows: 0 min, 10% B; 1–24 min, 24% B; 24–28 min, 32% B; 28–36 min, 40% B; 36–40 min, 60% B; 40–47 min, 95% B; 47–50 min 10% B; and 50–55 min, 10% B. The injection volume was 20 μL. The ionization conditions were a capillary temperature of 350 °C and a voltage of 4 kV. The nitrogen flow rate was 11 L/min, and the nebulizer pressure was 65.0 psi. The full-scan masses covered the range from m/z 100 to m/z 1200. Helium was used as a colliding gas in the ion trap, with voltage ramping cycles from 0.3 to 2.0 V.

### 4.7. Gas Chromatography/Mass Spectrometry (GC/MS) Analysis

The acquisition of chromatograms and mass spectra was performed using an Agilent Technologies 7890 gas chromatograph (Wilmington, Delaware, USA) equipped with an ALS 7683B autosampler and MSD 5975C mass selective detector operating in electron ionization (EI) mode at 70 eV. The carrier gas used was Helium UAP 5.0 at 1 mL min^−1^. The column used was a Zebron ZB 5Msi^®^ (30 m × 0.25 mm x 0.25 μm), and the system was calibrated with a C7–C40 n-alkane series (Sigma Aldrich 49452U). The temperature program for all separations was 2 min at 60 °C and 60 °C–260 °C at 5 °C/min. The injector temperature was 280 °C in splitless mode. Acquisition of spectra and chromatograms (SCAN) was performed using a single quadrupole mass selective detector (sQUAD) programmed at a temperature of 150 °C. The detector was tuned in automatic mode throughout all experiments, and the mass acquisition ranges were selected in scan mode for metabolites (45–850 m/z).

GC/MS analysis was performed on the essential oil and on extracts of *M. spicata* obtained with different solvents. For the quantification of the major compound carvone, the external standard calibration method was used. Subsequently, 2 µL of essential oil was dissolved in 998 µL of hexane (dilution 1:500). The essential oil content was determined from the relative percentage of the compounds and the yield percentage of the essential oil [41]. For the analysis of terpenoid compounds in the samples, the evaluation of apolar solvent extractions and extraction with ethanol at 80% was carried out to obtain the terpenoids and determine their relative percentage compared to the essential oil. One microliter of the samples was injected into the gas chromatograph. The data obtained were compared with the commercial National Institute of Standards and Technology (NIST) 2017 library, accepting a score greater than 70%. Agilent Masshunter Qualitative software was used for automatic deconvolution of the spectra and signal processing. The proportion of each individual compound was expressed in concentration as its relative percentage over the total percentage of compounds in the sample.

### 4.8. Kinetics of Solid–Liquid Extraction by High-Intensity Ultrasound

In the development of the high-intensity ultrasound (HIU) extraction process, the equipment used was an ultrasonic liquid processor LSP-500 (Sonomechanics, New York, NY, USA) provided with a 500-W ultrasonic generator, an air-cooled piezoelectric transducer (ATC-500), a full wave Barbell HornTM (FBH, 21 mm diameter tip), and a reactor chamber (304 stainless steel). A Masterflex L/S peristaltic pump with an Easy-Load II pump head was used for the continuous mode: a 1 L glass-sleeved vessel and circulating cooling bath (cooling temperature 10 °C) [37]. The experiments were performed using a polynomial regression model and Peleg’s model. The modified Peleg equation in the case of extraction is:(1)$$C_{\left(t\right)=}\frac{t}{K_{1}+K_{2}t}$$
where C_(t)_ is the concentration of rosmarinic acid, carvone, or total polyphenols at time (mg RA/100 mL; mg carvone/100 mL and mg equivalents of gallic acid (EAG)/100 mL); t is the extraction time; and K_1_ is Peleg´s rate constant (min × 100 mL/mg). The Peleg rate constant K_1_ is associated with the extraction rate (V_i_) at the beginning (t = t0):(2)$$V_{i}=\frac{1}{K_{1}}\;\left(\frac{\mathrm{mg}}{\min}\times 100\mathrm{mL}\right)$$

K_2_ is Peleg’s capacity constant (100 mL × mg) with which the maximum extraction yield and equilibrium concentration of rosmarinic acid, carvone or TPC (C_e_) can be determined when t→∞.
(3)$$C_{t\to \infty }={\;C}_{e}=\frac{1}{K_{2}}\;\left(\mathrm{mg}/100\mathrm{mL}\right)$$

The theoretical productivity of the process was calculated using Equation (1) with some modification:(4)$$t_{=}\frac{C_{\left(t\right)}K_{1}}{1-C_{\left(t\right)}K_{2}}\;$$
(5)$$P=\frac{V}{t}$$

The polynomial regression model was proposed to evaluate the effect of the extraction time and amplitude factors. Three levels of amplitude (40, 65, and 90) and 9 levels for time (2, 4, 6, 8, 10, 15, 20, 25, and 30) were evaluated to obtain the compounds of interest. As a response, the TPC, RA, and carvone were expressed in terms of mg EAG, mg rosmarinic acid and mg of carvone per 100 mL of extract, respectively. Response surface and contour plots were acquired using:(6)$$Y={\;\beta }_{0}+\beta_{A}{*\;X}_{A}+\beta_{B}{*\;X}_{B}+\beta_{\mathrm{AA}}{\left(X_{A}\right)}^{2}+\beta_{\mathrm{AB}}X_{A}X_{B}+\beta_{\mathrm{BB}}{\left(X_{B}\right)}^{2}$$
where *Y* is the dependent variable (TPC, RA, or carvone); β represents each of the regression coefficients; the subscripts _0_, _A_, _B_, _AB_, _AA_, and _BB_ represent the intercept, linear, quadratic, and interaction terms between factors, respectively; and X_A_ and X_B_ represent the independent values of each variable (factor). Nonlinear regression analysis (Peleg and polynomial model) was performed using Statgraphics Centurion version XVI (Statpoint Technologies, Inc.; Warrenton, VA, USA). Kinetic graphs and single-factor analysis were executed by GraphPad Prism^®^ version 5.00 software for Windows (GraphPad Software, Inc.—San Diego, CA, USA, 2007). Analysis of variance (ANOVA) was performed for each variable to test statistical significance using a *p* value at the 5% level.

## 5. Conclusions

This work shows the first chemical comparison study between byproducts and exportation material in *M. spicata*. Marketing byproducts are the best material compared to export material for use in different pharmaceutical, cosmetic, and food industries because these byproducts have the higher content of phenolic compounds, with twenty-seven compounds of this type tentatively identified. Marketing byproducts also had the highest amount of carvone. However, the total relative percentage of terpenoids was similar between export material and marketing byproducts. The comparative analysis between the different growth conditions indicates that the samples of *M. spicata* that presented higher phenolic compounds correspond to those cultivated in the open field, presenting significant differences with the samples cultivated under the greenhouse conditions. This work also shows the first simultaneous extraction by HIU for phenolic and terpenoid compounds using the same solvent system (ethanol:water 80:20). Through the design of experiments, the extraction process was standardized, minimizing the use of conventional technologies that generate high energy costs and are highly polluting. Proposing a reproducible, efficient, and potentially scalable technique to improve the extractions was therefore possible.

## Figures and Tables

**Figure 1 molecules-27-02243-f001:**
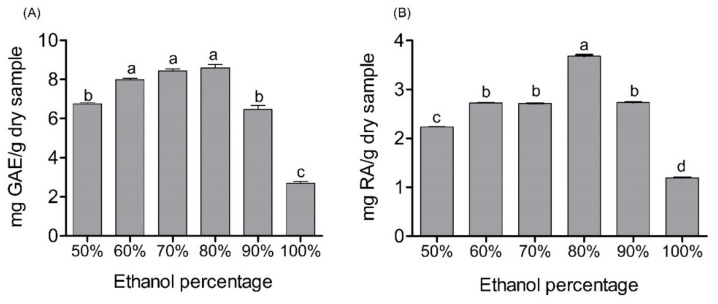
Selection of extraction system. Total polyphenols (TPC) (**A**), and rosmarinic acid content (RA) (**B**), in extracts obtained with different proportions of solvent (mg of the compound or mg of gallic acid equivalents (GAE) per gram of dry sample or extract). Statistical analysis uses a one-way ANOVA (Bonferroni test, *p* < 0.05). Equal letters means that there is no statistically significant difference.

**Figure 2 molecules-27-02243-f002:**
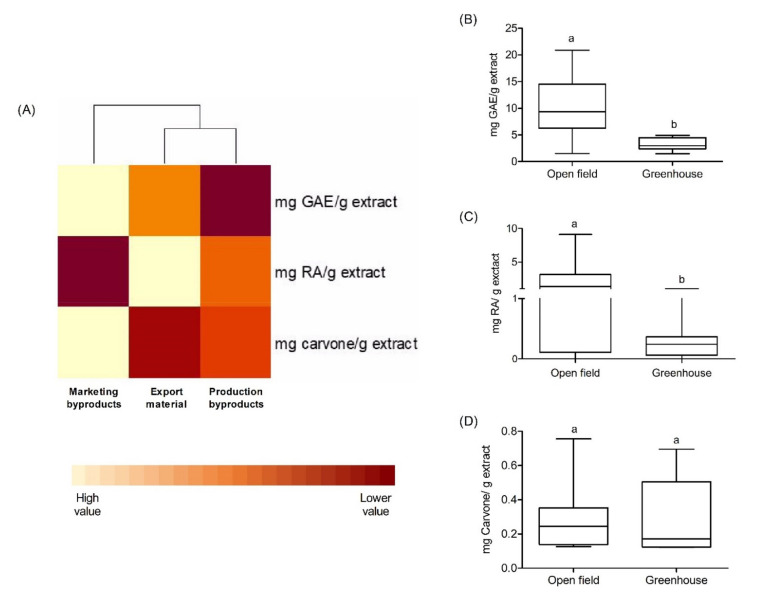
Heatmap of comparative analysis of byproducts and export material in the expression of GAE, RA and carvone (**A**). Box and whisker plots of the comparison of the open field and greenhouse in relation to the total phenolic content (**B**), rosmarinic acid (**C**), and carvone (**D**). Statistical analysis used one-way ANOVA (Bonferroni test, *p* < 0.05). Equal letters indicate that there is no statistically significant difference.

**Figure 3 molecules-27-02243-f003:**
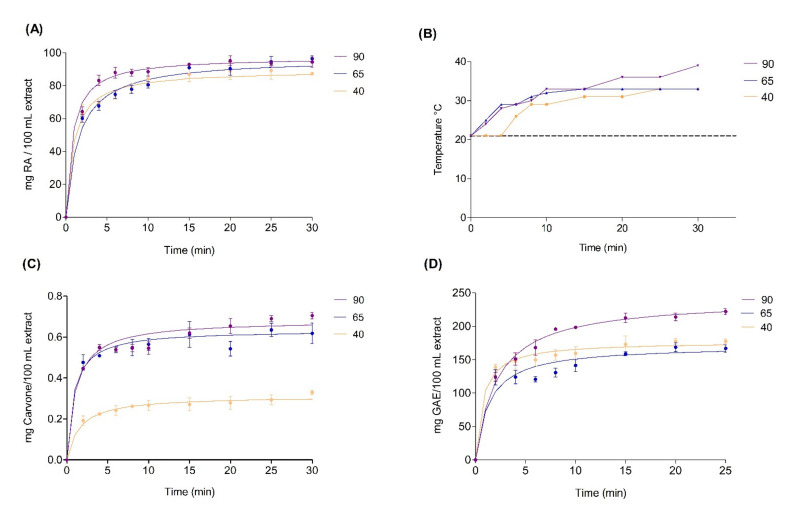
Kinetics extractions of total polyphenols (TPC) (**A**), temperature changes during the extraction at different temperatures (**B**), kinetics extractions of rosmarinic acid (**C**), and kinetics extractions of carvone (**D**).

**Figure 4 molecules-27-02243-f004:**
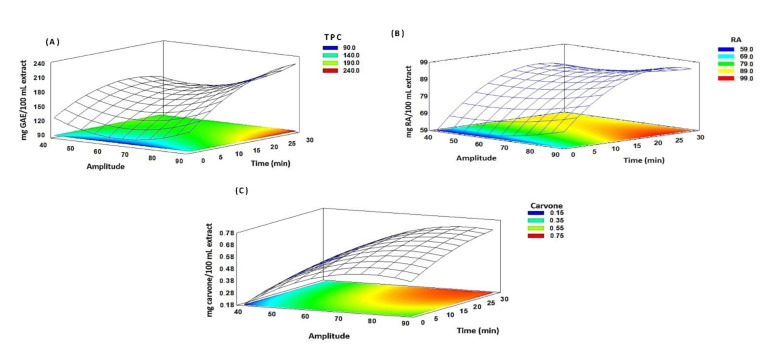
Response surface and contour plot effect of the extraction and amplitude for total polyphenols (TPC) (**A**), rosmarinic acid content (RA) (**B**), and carvone (**C**).

**Table 1 molecules-27-02243-t001:** Chemical composition of the different *M. spicata* sample extraction systems.

Extraction Solvents % Relative							
Compound	Essential Oil	Hexane: Ethyl Acetate 50:50	Hexane: Ethyl Acetate 75:25	100% Hexane	100% Ethyl Acetate	100% Dichloromethane	80% Ethanol
D-Limonene	4.28						0.23
Neodihydrocarveol	0.61						
Trans-Carveol	4.00	3.72	2.82	9.11	6.09		
Carvone	58.52	17.89	12.64	39.42	26.36	25.85	30.51
Dihydrocarvyl acetate				0.57			
Trans-Carveyl acetate				0.75			
γ-Elemene				0.67			
β-bourbenene	3.59			0.79			3.11
β-Elemene	1.75						0.47
β-Caryophyllene	4.55						8.96
β-Copaen-4α-ol	1.17				3.06		
ε-Muurolene							2.54
(-)-Isogermacrene D	0.84			4.10			2.39
(E)-β-Famesene	0.93						1.5
cis-Muurola-4(15),5-diene	1.07			1.60	0.80		5.72
Germacrene D	2.55	2.24	0.78	2.99	1.66	6.23	6.26
γ-cadinene							
Trans-Calamenene	0.69						3.24
Monoterpenoids	67.41	21.61	15.46	49.85	32.45	25.85	30.74
Sesquiterpenoids	17.14	2.24	0.78	10.15	5.52	6.23	34.19
TOTAL	84.55	23.85	16.24	60.00	37.97	32.08	64.93

The relative percentage was based on 100% of the compounds, but only the identified compounds are reported in the table.

**Table 2 molecules-27-02243-t002:** Main compounds in *M. spicata* tentatively identified by HPLC-DAD-ESI/MS^n^.

Peak	RT (min)	[M-H] m/z	MS^n^ (MS^2^, MS^3^) Experimental	Tentative
1	6.5	353	191, 179, 135	Neochlorogenic acid
2	8.5	311	179, 149	Caftaric acid
3	9.0	337	163, 119	3-ρ-coumaroyl-QA
4	9.5	325	163, 119	ρ-coumaryol-hex
5	10.0	353	179, 173	Criptochlorogenic acid
6	12.3	179	159, 153	Caffeic acid
7	12.8	593	473, 383, 503, 353, 297	Vicenin 2
8	13.7	337	173, 137, 111	4-ρ-coumaryol-hex
9	14.7	377	359, 265	Rosmarinic acid derivative I
10	15.0	377	359, 197, 135	Rosmarinic acid derivative II
11	15.2	367	191, 173	4-Feruloyl-QA
12	18.9	595	287, 151	Eriocitrin
13	20.0	593	285, 241, 175, 151	Luteolin-7-*O*-rutinoside
14	21.4	447	285	Luteolin-7-*O*-glucoside
15	21.8	461	285, 243, 175	Luteolin-7-*O*-glucuronide
16	23.3	579	271	Narinrutin
17	24.3	577	269, 225	Apigenin-7-*O*-rutinoside
18	24.7	717	519, 475, 365, 321	Salvianolic acid
19	25.7	609	301, 286, 227	Hesperidin
20	27.0	445	269, 225, 175	Apigenin-7-*O*-glucuronide
21	27.5	359	359, 223, 197, 161	Rosmarinic acid
22	29.0	461	315, 285, 241	Kaempferol-7-*O*-glucuronide
23	30.9	537	493, 359, 295	Lithospermic acid
24	31.7	533	387, 369, 207, 163	Medioresinol-*O*-rhmanoside
25	32.2	563	387, 370/369, 207	Medioresinol-*O*-glucuronide
26	33.4	493	359, 223, 179, 161	Rosmarinic acid derivative III
27	34.1	493	359, 223, 197, 179, 161	Rosmarinic acid derivative IV

**Table 3 molecules-27-02243-t003:** Comparison of export material and byproducts in relation to terpenoids commonly present in the essential oil of spearmint.

Compound	% Relative			
	Essential Oil	Export Material	Production Byproducts	Marketing Byproducts
D-Limonene	4.28	1.22	1.37	1.91
Neodihydrocarveol	0.61	0.43	0.87	4.08
trans-Carveol	4.00	3.15	2.41	3.11
Carvone	58.52	13.25	12.47	8.91
Dihydrocarvyl acetate				0.72
trans-Carveyl acetate		0.28		0.68
γ-Elemene		0.25	0.19	
β-bourbenene	3.59	1.30	1.58	1.76
β-Elemene	1.75	0.35	0.40	
β-Caryophyllene	4.55	1.93	1.93	1.49
β-Copaen-4α-ol	1.17	0.26	0.47	
ε-Muurolene		0.39	0.32	
(-)-Isogermacrene D	0.84	0.26	0.35	
(E)-β-Famesene	0.93	0.42	0.45	
cis-Muurola-4(15),5-diene	1.07	0.35	0.41	0.40
Germacrene D	2.55	2.89	2.63	1.46
γ-Elemene		0.42		
trans-Calamenene	0.69	0.27	0.27	
(-)-Spathulenol		0.28	0.08	
α-Cadinol		0.49	0.97	2.26
(1R,7S,E)-7-Isopropyl-4,10-dimethylenecyclodec-5-enol		0.36	0.79	0.78
TOTAL	84.55	28.54	25.54	27.56

## Data Availability

Not applicable.

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
