# Peer review of "Spearmint (Mentha spicata L.) Phytochemical Profile: Impact of Pre/Post-Harvest Processing and Extractive Recovery"

_molecules, 2022, doi:10.3390/molecules27072243_

Round 1
Reviewer 1 Report
The authors aimed to i) Select the best solvent-system for efficient recovery of polar (phenolic compounds, PC) and non-polar (terpenoids) phytochemicals from spearmint (Mentha spicata L.) leaves, ii) Compare the polar/non-polar phytochemical profile of different samples [post-harvest byproduct (PHB), export material (EM), marketing byproducts (MB); open filed (OPF) vs. greenhouse (GRH)], and iii) Optimize a high-intensity ultrasound (HIU; variables: time and amplitude) method to extract phytochemicals from the assayed samples. 80% ethanol was the best extractive solvent for PC, rosmarinic acid (RA), and terpenoids (mainly carvone, trans-carveol, β-186 caryophyllene, and germacrene, OPF>GRH / EM> PHB/MB were the richest sources of PC and RA but MB was for terpenoids, and 90% amplitude for five minutes was the best HIU condition for terpenoids and PC.
Although findings are very relevant, the experimental design and the way results are reported and discussed, seem to reflect independent (not consecutive) experiments. This is due, in part, to a poor English translation and the fact that all variables & combinations [3 samples (EM, PHB, MB) x growing conditions (OPF, GRH) = 6 experimental treatments) are not systematically included in all tables/figures and the whole result section. To improve the scientific soundness and experimental design, the authors are asked to consider the following:
General
- The readability and syntax of the manuscript will be substantially improved if it is reviewed by a formal translation agency or by a native English spoken person.
- Include a flowchart (possibly as supplementary material) specifying how the analyzed variables & combinations [3 samples (EM, PHB, MB) x growing conditions (OPF, GRH) = 6 experimental treatments) were included in each of the experimental stages and explain within text why any of them was not included at a given step.
Sections
- Title. Quite long. Suggestion: Spearmint (Mentha spicata) phytochemical profile: Impact of pre/post-harvest processing and extractive recovery.
- Abstract. Once the changes in the results section (see below) are considered, this section should be more quantitative (including p-values) reflecting best vs. best. worst sources of phytochemicals and optimized extraction conditions.
- Introduction. Authors should review from this section onwards how references are reported (e.g. line 70 Chang et al. and Chemat et al. are references 13, 14).
- Results/discussion. -
- Methods. Section 4.2: A more detailed discussion of samples (EM, PHB, MB), growing conditions (OPF, GRH), and their combinations (n=6?) analyzed in this study, should be included.
- Tables. Table 2 can be included as supplementary material and instead include a new table reflecting differences in composition between experimental treatments as you did in Table 3. HIU kinetic equations should be included in a new table or as part of Figures 4a-c.
- Figures. Improve the resolution of all figures (≥300 dpi) and highlight statistical differences between all treatments from higher-to-lower values [e.g., Fig 1: 50-100% EtOH treatments could be b-a-a-a-b-c (PC) and c-b-b-a-b-d (RA), respectively]. Authors should depict and explain within the text the effect of temperature (Figure 3), amplitude, and time (Figure 4) as synergistic variables, if possible in the same surface plots.
- Conclusion. Change if necessary, according to the new suggestions
- References. Authors must reduce the number of references≥ 10y old to ≤20% and references’ format should comply with Molecules´ guidelines.
Reviewer 2 Report
The paper deals with an interesting application of experimental design to the optimization of the extraction of terpenoids and phenolic compounds from Metha Spicata samples. The paper is interesting and original, however, some major revisions are needed in order to make the paper and the results clearer to the readers and to point out the main novelties. Moreover, English should be revised throughout the paper to make it clearer.
Here, the more general comments and questions are listed:
- Introduction: the authors should highlight the main novelties of the work, together with the objectives that should be indicated more clearly.
- I was not able to find Figures S1 and S2. I imagine theu are in the Supplementary Material but I did not find it in the Download area. Also Tables S1-3 are missing.
- Samples: it is not clear how many samples were analyzed for each farm and if the same number of samples was available for each farm bot in natural conditions and grown in a greenhouse. How were the samples selected? (e.g., hw were the samples selected in the field? From the opposite sides of the same field, randomly etc?)
- Table 1 reports the results obtained with different extraction solvents: it is not clear why mixtures hexane:ethylacetate were compared to pure dichloromethane. Since the authors exploit experimental design, I would have expected them to exploit mixture designs to explore different mixtures of at least three components (hexane, ethyl acetate and dichloromethane). Similarly, it is not clear why ethanol was used first and then compared to mixtures with other solvents. As the authors state in line, 101-103, ethanol is approved for use in the food industry. If it is so, why the exploration of other solvents? I think the authors should present the results in a clearer way, to help the reader understand the steps. Moreover, in Table 1 and also in the other tables, it is not clear how the relative % are calculated: are they calculated as the signal of each analyte divided by the total signal? If it is so, why they do not clore to 100%? Please give details.
- Figure 2A: do the three types of samples correspond to the evarage values obtained? Usually heat maps are provided for single measurements. Moreover, it is not clear if these results include both the samples grown in open fields and in greenhouses. I think it would be better to separate them and provide the heat map for single measurements.
- HPLS-DAD-ESI-MS: in line 172 the authors say that the identification of the analytes was done by database search. How was this step performed? It is quite challenging to use database search in HPLC applications, please provide details about the procedure used for analytes identification.
- Table 2: please give details about why some of the masses are in bold. I could not find it in the text. It would be better to explain it also in the Table caption.
- Table 3: why the results are presented separately for the three types of sample but not for the two types of growing conditions? I think both information should be presented? Are these results calculated averaging the two different growing conditions? If it is so, they shuld be provided separately.
- Line 200: the authors list three paramters that should be used in HIU, however, they did not provide a description of these parameters and moreover, the parameters undergoing experimental design are not the same.
- Experimental design. The authors explain they exploited experimental design techniques. Since it is not very usual to find applications of experimental design to optimization of lab procedrures, I think the authors should highlight this fact more clearly in the paper. I think they could add a paragraph 4.9 just about experimental design, highlighting clearly which parameters were evaluated, in which range and which experimental design was adopted and with which repetitions (the centre of the domain? With repetitions of all the experiments?) When the authors speak about amplitude, it is not clear what they refer to. I think they should briefly explain the parameters that can be varied in the extraction procedure. Since experimental design was adopted, I expected to find the final model described, together to the ANOVA table of the effects. It is also not very clear to me if the authors applied the same design of experiments to all the mixtures of just to 80% ethanol. If this last alternative is the right one, please explain your choices.
- Figure 4: the figures are not very clear, please rotate them to make the surfaces clearer or provide 2D contour plots instead. The interaction could also be commented in more detail.
- Chromatographic Methods: in paragraphs 4.5, 4.6 and 4.7 it is not clear if the methods were validated and if matrix effects were detected or not.
- Paragraph 4.7: please, add the details about the mass analyzer and the range mass exploited.
- Line 361: the authors describe how essential oil was obtained. I think this information should be moved in a paragraph dedicated to samples description. Please, also provide reasons for the use of esential oil as a sort of reference. It is clear to me but perhaps it could be not so clear to a wider platea of readers.
- Conclusions: I find the conclusions very poor. I think the authors should expand them to highlight the novelty of the work and the major important findings, together with their impact on the field. Moreover, the conclusions in their present form are quite difficult to understand due to English use.
- English should be improved in the overall paper.
Minor comments are provided hereafter:
- Line 18: recovery of the compounds
- Line 20: The maximum amount of compounds was obtained
- Line 21: thanol/water mixture (80:20)
- Line 23: monoterpenoid and sesquiterpenoid compounds
- Line 24: were higher
- Line 25-26: The sentence is not clear.
- Line 34: wastes, duing the production processes exploiting roots
- Line 53: monoterpenoid compounds
- Line 53: phenolic compound corresponds
- Line 69: change rupture with breaking
- Line 69: eliminate of the particles
- Line 72: in the industry of phytotherapeutic products
- Line 75: which is more difficult with other
- Line 76: to compare the content of phenolic
- Line 78: obtained from crops cultivated in the open field or under
- Line 81-84: the sentence is not clear
- Line 100: the last mixture was selected
- Line 105: phenolic compounds, terpenoids commonly
- Line 106: also extracted with the chosen solvent
- Line 107: objective was
- Line 107: this type of compounds
- Figure 1 and in all the paper: are these amounts over g of dry sample?
- Line 131: during food storage
- Line 133: ranged from 6.40 to
- Line 140: by the environmental conditions
- Line 152: per dry weight
- Line 152: factors related to the growth, the environment and the crop conditions.
- Line 175-176: the sentence is not clear
- Line 187: majority of the extracted compounds. Extraction with solvents
- Line 218: reaching a plateau
- Line 208-226: plese take care of all superscripts when R2 is indicated
- Line 228-239: please use the subscripts where X1, X2 etc are indicated and use superscripts where R2 is indicated
- Line 311: use subscripts correctly in the formula
- Line 316: is it dry sample?
- Line 337: was a Phenomenex
- Line 337: I think it is 5 µm particle size
- Paragraph 4.8: please use correctly superscripts and/or subsripts in the equation and in the description of the parameters included in the equation
- Line 403-411: please use correctly superscripts and/or subsripts in the equation and in the description of the parameters included in the equation
Reviewer 3 Report
This manuscript compares the chemical differences between two sets of compounds in M. Spicata grown in green house and also in the open field. Authors have used the well known HIU extraction to answer their hypothesis. They have used HPLC and also the GCMS to identify and validate the data.
The paper reads very well and the story has been told in the perfec way. authors have considered/used all the required experiments to draw their conclusions. Figures and tables are flawless and very informative. i got most of their message by seeing their figures and also tables. i am happy for the article to publish as it is.
Reviewer 4 Report
Dear Authors
Please fix these issues:
1- Increase the resolution of all figures.
2- Please use italic for M. spicata in whole manuscript
3- What was the yield of extraction using the 6 mixtures established in this work
Good Luck
Round 2
Reviewer 1 Report
Thank you very much for having accepted most of my suggestions, the manuscript has improved substantially
Reviewer 2 Report
The paper can be accepted in its present form
Reviewer 4 Report
Dear authors
Special thanks for the improvements done.
Good luck